

# Restoring South African subtropical succulent thicket using *Portulacaria afra*: exploring the rooting window hypothesis

Nicholas C. Galuszynski[1,2], Ryan E. Forbes[3], Gavin M. Rishworth[4,5] and Alastair J. Potts[1,2]

[1] Spekboom Restoration Research Group, Nelson Mandela University, Gqeberha, South Africa
[2] Botany Department, Nelson Mandela University, Gqeberha, South Africa
[3] Centre for African Conservation Ecology, Zoology Department, Nelson Mandela University, Gqeberha, South Africa
[4] Institute for Coastal and Marine Research, Nelson Mandela University, Gqeberha, South Africa
[5] Zoology Department, Nelson Mandela University, Gqeberha, South Africa

Corresponding author
Alastair J. Potts,
alastair.potts@mandela.ac.za,
potts.a@gmail.com

## ABSTRACT

Drought prone, arid and semi-arid ecosystems are challenging to restore once degraded due to low levels of natural recruitment and survival of reintroduced plants. This is evident in the restoration of degraded succulent thicket habitats in the Albany Subtropical Thicket Biome located in South Africa. The current restoration practice for this ecosystem focuses predominantly on reintroducing *Portulacaria afra* L. Jacq., which is naturally dominant in terms of cover and biomass, but largely absent in regions degraded by domestic livestock. This has been achieved by planting unrooted cuttings with limited consideration of soil water availability in a drought-prone ecosystem. This study tests the effects of the timing of water availability after planting on the root development of *P. afra* cuttings. Cuttings were harvested from seven individual plants and grown in a glasshouse setting. Eighty four cuttings were taken from each individual, twelve for each of the seven watering treatments per individual plant. The treatments represented a time-staggered initial watering after planting, including: on the day of planting, 4 days, 7 days, 14 days, 21 days, and 28 days after planting. After 32 days, all treatments were watered on a bi-weekly basis for two weeks; a control treatment with no watering throughout the experiment was included. The proportion of rooted cuttings per treatment and dry root mass were determined at the end of the experimental period (day 42). The early onset of watering was associated with a higher percentage of rooting ($X^2(5) = 11.352$, $p = 0.045$) and had a weak, but non-significant, impact on the final dry root mass ($F_{5,36} = 2.109$, $p = 0.0631$). Importantly, no clear rooting window within 28 days was detected as the majority of cuttings exhibited root development (greater than 50% of cuttings rooted for each individual parent-plant); this suggests that watering at the time of planting *P. afra* cuttings in-field for restoration may not be necessary. An unexpected, but important, result was that parent-plant identity had a strong interaction with the accumulation of root mass ($F_{36,460} = 5.026$, $p < 0.001$; $LR_7 = 122.99$, $p < 0.001$). The control treatment, which had no water throughout the experiment, had no root development. These findings suggest that water availability is required for the onset of rooting in *P. afra* cutting. However, the duration of the experiment was insufficient to detect the point at which *P. afra* cuttings could no longer initiate rooting once exposed to soil moisture, and thus no rooting window could be

---

defined. Despite harvesting material from the same source population, parent-plant identity strongly impacted root development. Further work is required to characterise the rooting window, and to explore the effect of parent-plant condition on in-field and experimental restoration results; we urge that experiments using *P. afra* closely track the parent-source at the individual level as this may be a factor that may have a major impact on results.

## INTRODUCTION

The persistence of arid and semiarid ecosystems in a degraded state is often maintained by complex interactions operating at various spatial and temporal scales (*D'Odorico et al., 2013*; *Evans & Geerken, 2004*). Restoration of these ecosystems may thus require targeting key processes, such as erosion, herbivory, pathogens, and drought (*James et al., 2013*). A lack of reliable rainfall, for example, can restrict the regeneration of degraded arid and semiarid ecosystems, resulting in low rates of plant establishment (*Valliere et al., 2019*; *Haase & Davis, 2017*). Extended periods of low soil moisture may contribute to the poor survival of *Portulacaria afra* L. Jacq. cuttings planted in succulent thicket restoration initiatives, especially since the standard protocol is to plant unrooted cuttings—a long period of low soil moisture may retard or prevent rooting.

Succulent thicket represents the arid and semiarid components of the Albany Subtropical Thicket biome (*Vlok, Euston-Brown & Cowling, 2003*), which is endemic to South Africa. Although this vegetation occurs in a region that experiences weakly seasonal rainfall, with peaks in spring and autumn, it is still subject to frequent multi-year droughts, including consecutive months with no rainfall (*Mahlalela et al., 2020*; *Palmer et al., 2020*; *Vlok, Euston-Brown & Cowling, 2003*). Despite this, succulent thicket has been described as an evergreen dwarf forest (*Midgley et al., 1997*), characterised by a low canopy of trees and shrubs—usually with a canopy cover > 70%—and an understory rich in succulents and geophytes (*Vlok, Euston-Brown & Cowling, 2003*). A common and often distinguishing characteristic of succulent thicket is the abundance of the leaf and stem succulent tree, *P. afra,* which is frequently the most dominant species in terms of canopy cover and biomass (*Guralnick & Gladsky, 2017*; *Vlok, Euston-Brown & Cowling, 2003*; *Penzhorn, Robbertse & Olivier, 1974*). This species plays a key role in landscape-level facilitation, modifying local environmental conditions through shading effects (*Wilman et al., 2014*; *Sigwela et al., 2009*), intercepting rainfall (*Cowling & Mills, 2011*), and improving water infiltration by accumulating soil organic matter (*van Luijk et al., 2013*; *Lechmere-Oertel et al., 2008*). This facilitation is purported to enable the persistence of a closed-canopy scrub forest-like vegetation in arid areas (*Lechmere-Oertel, Kerley & Cowling, 2005b*).

Livestock production has caused widespread and extreme degradation of succulent thicket vegetation, which exhibits limited evidence of natural regeneration (*Sigwela et al., 2009*; *Lechmere-Oertel, Kerley & Cowling, 2005b*). This degradation is characterised by

the complete loss of *P. afra* and the associated transition into a savanna-like habitat with limited ecological functioning (*Lechmere-Oertel, Kerley & Cowling, 2005b*; *Lechmere-Oertel, Cowling & Kerley, 2005a*; *Lechmere-Oertel et al., 2008*). Succulent thicket restoration, therefore, focuses predominantly on the reintroduction of *P. afra* as both a pioneer and ecosystem-engineering species.

The ability of *P. afra* to regenerate clonally is well documented (*Oakes, 1973*; *MacOwens, 1897*) and may reflect a co-evolution with elephant browsing (*Stuart-Hill, 1992*). This trait has been exploited in succulent thicket restoration programs as unrooted *P. afra* cuttings have established after being planted across a wide range of degraded thicket sites (*Van der Vyver, Mills & Cowling, 2021a*; *Mills & Robson, 2017*; *Mills et al., 2015*). The simplicity of *P. afra* reintroduction and potential for restored habitats to act as carbon sinks (*Van der Vyver & Cowling, 2019*; *Mills & Cowling, 2006*; *Mills & Cowling, 2014*) prompted investigations into the feasibility of succulent thicket restoration at a biome scale (*Mills et al., 2015*): termed the "Thicket Wide Plot (TWP)" experiment. A total of 330 restoration plots were established across the natural range of succulent thicket between 2008 and 2009; each plot comprising a $50 \times 50$ m herbivore exclosure within which various unrooted planting treatments were tested (described in *Van der Vyver, Mills & Cowling, 2021a*). However, the survival within these plots ranged from zero to nearly 100% (*Van der Vyver et al., 2021b*). The factors responsible for the low survival include frost, herbivory, and planting outside of the target habitat (*Duker et al., 2020*; *Van der Vyver et al., 2021b*), although the influence of weather conditions prior to and post planting could not be explored due to a lack of data.

As mentioned above, succulent thicket occurs within weakly seasonal rainfall zone, receiving sporadic rainfall predominantly in autumn and spring, with an annual mean ranging from 100 to 450 mm (*Vlok, Euston-Brown & Cowling, 2003*; *Everard, 1987*) and prolonged droughts are common with variable local rainfall patterns. Thus, the soil moisture necessary to stimulate and support root growth may be absent for periods longer than the survival of unrooted cuttings. Here we hypothesise that soil moisture in the first month after planting may affect the initiation of rooting and thus root growth of this succulent species, potentially providing insights into the variable survival reported in field plantings (*Van der Vyver et al., 2021b*; *Mills & Robson, 2017*).

## METHODS

### Harvesting

Cuttings were harvested on 27 July 2020 from a road verge within the city of Gqeberha, Eastern Cape, South Africa, where *P. afra* had been used to stabilise a road verge—*i.e.*, used for slope rehabilitation; this experiment was conducted during the 2020 COVID-19 pandemic, where movement was highly restricted and thus material available for sampling was geographically limited. Eighty-four cuttings were harvested from seven randomly selected individual plants (hereafter "parent-plants"). Twelve replicate branchlets per parent-plant were used in each of the seven watering treatments (described below). The stem length and basal diameter were recorded for each cutting prior to planting; the

cuttings were smaller than those used in the standard planting protocol or the TWP experiment (also termed "truncheons" in *Van der Vyver, Mills & Cowling, 2021a*; *Van der Vyver et al., 2021b*), with a stem width mean ± sd of 5.60 ± 0.87 mm and a cutting length of 222 ± 26 mm. Smaller branches were used (hence cutting) to accommodate the limited space available in seedling trays and the glasshouse area.

## Experimental layout

A total of twelve seedling trays, with planting cavities of 90 ml, were used for the experiment. All planting cavities were filled with clay-rich soil obtained from an area supporting thicket. This soil was standardised by drying it in an oven at 60 °C until it reached a constant mass, then sieving it through a one-millimetre mesh to homogenise soil particle size.

All planting was conducted on 28 July 2020. Each tray included seven treatment rows, with one cutting per parent-plant represented in each treatment. Each row was separated by an empty cavity to reduce shading effects and potential water overflow between adjacent treatments; also, all trays were raised to avoid water entering from the bottom of a cavity (see Fig. S1 for more details). Treatment order was randomised across each tray, and parent-plant order was randomised in each treatment by row. All trays were placed in a well lit portion of a glass house and randomly repositioned and rotated on a weekly basis. Maximum and minimum temperature, and relative humidity were monitored in the glass house for the duration of the experiment (see Fig. S2 for further details).

## Watering treatments

The watering treatments represent a time staggered initial watering after planting. This included watering on the day of planting (D01), 4 days (D04), 7 days (D07), 14 days (D14), 21 days (D21), and 28 days (D28) after planting, and a control treatment (C) with no watering for the full duration of the experiment. After 32 days from the start of the experiment, all treatments were watered twice weekly for an additional two weeks. All watering events involved saturating the soil.

## Data collection and analysis

After 42 days from initial planting, all cuttings were carefully removed from the soil and the percentage of cuttings per parent plant that rooted per treatment was calculated. The roots were removed, and the soil was searched to ensure that all roots and root fragments were accounted for. Each cutting's roots were rinsed to remove any excess soil and subsequently dried at 60 °C until constant weight. Dry root mass was measured in grams to the third decimal-place.

All analyses described below were performed in R v 4.2.1 (*R Core Team, 2022*). To test the rooting window hypothesis, differences in rooting success (percentage rooted cuttings within each plant across treatments) was evaluated *via* a nonparametric Kruskal-Wallace test (*Hollander & Wolfe, 1973*). The dry root mass produced during the experimental period was evaluated and compared using three approaches. Firstly, using a nested ANOVA approach ("RootMass~Treatment/Plant"; *Chambers, Freeny & Heiberger, 1992*) coupled with a post-hoc Tukey test (*Yandell, 1997*). Secondly, Kruskal-Wallis tests were performed on the dry root mass across all treatments, and then also separating the data into individual

parent plants across all treatments; *post-hoc* analyses were conducted on sets that returned < 0.05 using the 'pgirmess' package (v 1.7.0; *Giraudoux, 2021*). The unwatered control treatment was not included in the Kruskal-Wallis tests or nested ANOVA described above as no rooting was observed in this treatment.

Additional trends in the data were explored following a more sophisticated approach using linear mixed-effects modelling (LMM) as implemented in the 'nlme' package (v 3.1-153; *Pinheiro et al., 2018*). To account for possible effects of stem width on root formation, as the larger area along the stem circumference of thicker stems may produce more roots, root mass was divided by stem circumference. The full fixed effects model was therefore represented as the standardised (x = 0; $\sigma = 1$) root mass divided by stem circumference as a function of treatment, or *RootMass/StemCirc ∼Treatment*. The optimal random structure was determined by comparing the residual fit of separate full models (*Zuur et al., 2009*; *Zuur, Ieno & Elphick, 2010*) that incorporated different combinations of either random intercepts per plant individual (random = ∼1|*Plant*) from which the cutting was taken or varying residual identity structure per plant (varIdent (form = ∼1|*Plant*)). These were compared under restricted maximum likelihood estimated (REML) using the log-likelihood ratio test and AIC scores (*Zuur et al., 2009*). The optimal residual variance structure included *Plant* individual as both a random intercept and identity structure. This model was then refit under maximum likelihood estimation to test if the removal of the fixed effect (*Treatment*) substantially reduced the information criterion (model fit) score, which it did not ($\Delta$AIC $\leq 2$). The same model was then refitted under REML estimation and the 'summary' statistics of the fixed effects per *Treatment* compared using the t-statistic. The marginal and conditional $R^2$ value for the most parsimonious model was calculated using the 'MuMIn' package (*v. 1.47.1*: *Nakagawa & Schielzeth, 2013*; *Barton, 2022*). All model assumptions in terms of homogeneity and normality of residuals were tested and met. The significance of statistical tests are reported according to the terminology proposed by *Muff et al. (2022)*.

## RESULTS AND DISCUSSION

### Rooting window hypothesis

The timing of initial watering was found to have a moderate significant effect on the proportion of *P. afra* cuttings that established roots (Fig. 1; $X^2$ (5) = 11.352, $p = 0.045$). However, the period with no watering was insufficient to detect the rooting window (*i.e.,* the point after which *P. afra* cuttings have very low likelihood of rooting). The total exclusion of watering (control) inhibited root initiation in all but one of 84 cuttings; this one cutting exhibited root tips and was considered not to have established roots. Early watering (within the first 4 days) resulted in the highest percentage of rooting across all parent-plants (Fig. 1). However, there was only weak evidence for differences in dry root mass amongst treatments by the end of the experiment ($F_{5,36} = 2.109$, $p = 0.063$; $X^2$ (5) = 9.719, $p = 0.084$; LMM: $F_{5,489} = 1.832$, $p = 0.105$), with some indication that root growth was maximised when watering began two to three weeks after planting (D14 and D21 in Fig. 2, and higher coefficients in the LMM, although not significant: Table 1). The
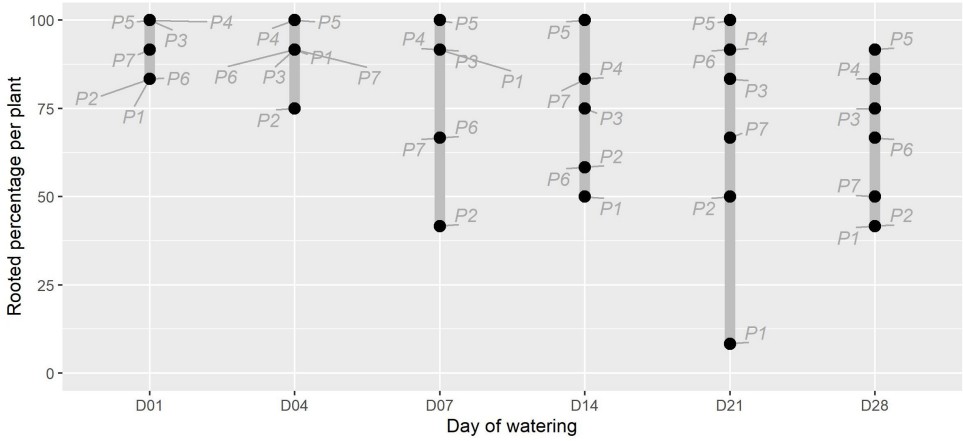

**Figure 1** **The percentage of rooted cuttings ($n = 12$) per plant (P1-P7) for each initial watering day (D01-D28).** The range for each watering treatment is indicated by the grey bar. There was moderate evidence that the watering treatments affected the proportion of rooted plants ($X^2(5) = 11.352$, $p = 0.045$). The parent plant number is indicated in grey text. Note that the unwatered control is excluded as no rooting was present in this treatment at the end of the experiment.

interaction terms with parent-plants are discussed separately in the section below entitled "Parent-plant effects". These preliminary findings may suggest that while the early onset of watering increased the success of root initiation, delayed watering might increase the relative accumulation of root mass in cuttings that do set root. Cuttings subject to delayed watering had a shorter period until the final four watering events that were applied to all treatments in the last two weeks (except the control)—this may have boosted root growth by avoiding moisture deficits at the early stages of root formation.

Root generation from stem cuttings consists of different processes: regeneration of damaged tissue at the wounding site, redifferentiation of cells to perform new functions (shoots to roots), root tip formation, and the elongation of root tissue to produce functioning roots (*Bidabadi & Jain, 2020*; *Cameron & Thomson, 1969*). These processes require different physiological conditions, and the outcome of cellular differentiation is influenced by the relative proportions of endogenous growth hormones (auxins promoting rooting initiation and cytokinins promoting root elongation) whose concentrations increase at the wound site in response to external stimuli (*Fehér, 2019*). The timing of watering *P. afra* cuttings may affect the abundance of these growth regulating hormones.

We were unable to find studies of growth regulating hormones in succulent species; however, water availability and rooting have been explored in woody C3 cuttings (reviewed in *De Almeida et al., 2017*; *da Costa et al., 2013*). In general, as cuttings are unable to take up moisture until they are rooted, water stress responses are initiated soon after the removal of cuttings from the parent-plant, with the wound site resulting in rapid moisture loss. Early watering of cuttings can facilitate the maintenance of a positive water balance, preventing desiccation and the upregulation of secondary metabolites, which can inhibit cell cycle progression (*Wolters & Jürgens, 2009*). The interaction of endogenous growth hormones
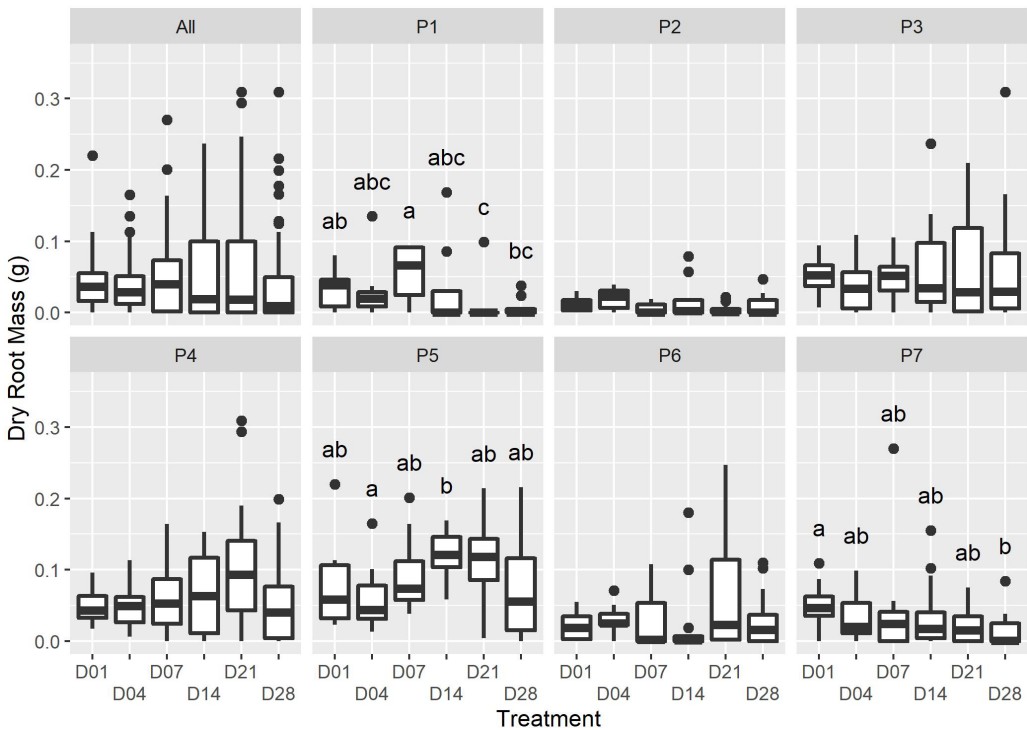

**Figure 2** **The dry root mass for each watering treatment across all plants ("All"), and each individual ("P1" to "P7").** Only minor differences in dry root mass were recorded amongst the treatments at the end of the experiment (ALL: $F_{5,36} = 2.109$, $p = 0.0631$; $X^2(5) = 9.719$, $p = 0.084$); however, there was strong evidence for an interactive effect of the individual on dry root mass ($F_{36,460} = 5.026$, P = $<2e-16$). Dissimilar letters indicate significant differences amongst treatments within parent-plant subsets (P1-P7; see Table S1 for Kruskal-Wallace test statistic values).

that regulate root initiation and growth may explain the abundance of slower-growing roots in early watered *P. afra* cuttings and the rapid growth of few roots in later watered cuttings (D21 and D28, Fig. 2; Table 1). However, further research is required to better understand the physiology of *P. afra* rooting and the effect of dry periods after rooting has been initiated.

## Parent-plant effects

An unexpected observation in the data is that the source of the cuttings, *i.e.,* the parent-plant, had a significant effect on the fraction of rooted cuttings and dry root mass (*e.g.,* P3, P4 & P5, Figs. 1 & 2; Table S1), providing strong evidence ($F_{36,460} = 5.026$, $p < 0.001$) for a parent-plant interaction effect on root growth. This is demonstrated when including individual *Plant* identity in the LMM model as a random effect significantly improved its parsimony compared to the null model ($df = 7$, LR = 122.99, $p < 0.001$) and also that the model $R^2$ fit when accounting only for the fixed effect (*Treatment*) was low (0.011) but substantially improved when the individual effect of *Plant* was included (0.219) (Table 1). The inclusion of both a random intercept and a residual variance structure per *Plant* identity within the most parsimonious model (all LR tests $p < 0.001$) suggests that there is

**Table 1** **Linear mixed-effects model of scaled root mass divided by stem circumference as a function of Treatment (day of watering commencement: D1-D28) fitted under restricted maximum likelihood estimation.** The reference category (D1) is compared to all subsequent treatments of watering commencement in terms of coefficients (C) and test (t; p) statistics. Plant identity (P1-P7) is accounted for as a random intercept (C) as well as a variable identity structure (var). DF = 489. The marginal $R^2$ (fixed effects only) was 0.011 while the conditional $R^2$ (fixed and random effects) was 0.219.

| Fixed effects | C (±SE) | t | p |
|---|---|---|---|
| *Intercept* | −0.09 (±0.21) | −0.43 | 0.67 |
| Treatment | – | 8.63[*] | 0.12[*] |
| D4 | −0.06 (±0.13) | −0.48 | 0.63 |
| D7 | 0.09 (±0.13) | 0.65 | 0.51 |
| D14 | 0.20 (±0.13) | 1.46 | 0.14 |
| D21 | 0.19 (±0.13) | 1.47 | 0.14 |
| D28 | −0.08 (±0.13) | −0.64 | 0.52 |

| Random effects | C | Var |
|---|---|---|
| P1 | −0.32 | 1 |
| P2 | −0.56 | 0.76 |
| P3 | 0.26 | 0.73 |
| P4 | 0.31 | 0.96 |
| P5 | 0.8 | 0.93 |
| P6 | −0.22 | 0.99 |
| P7 | −0.26 | 1.1 |

Notes.
[*]The test significance of Treatment overall is calculated using the log-likelihood ratio scores under maximum likelihood estimation.

both a parent-plant specific difference in the overall dry root mass accumulated per cutting and a difference in the variability between cuttings from a single parent-plant in terms of overall dry root mass accumulated (*i.e.,* some plants were more consistent or more variable in the dry root mass accumulated per cutting than others). As the experimental design did not intend to describe parent-plant effects on rooting, we did not measure any in-field attributes of the parent-plants at the time of harvesting. Therefore, the discussion of this is speculative, but important, as tracking parent source material has not been included in the design of any experiments exploring *P. afra* establishment thus far (including the TWP experiment). Thus, it may be a hidden but significant confounding factor in such experiments and requires further research and consideration in experimental design.

Cellular redifferentiation and growth require living material with sufficient internal resource supply for cell division. The rooting success of cuttings is potentially affected by tissue age (studied in *Diploknema butyracea*: *Zargar & Kumar, 2018*; *Dalbergia melanoxylon*: *Amri et al., 2010*; *Tectona grandis*: *Husen & Pal, 2007a*; *Husen & Pal, 2007b* *Quercus spp.*: *Chalupa, 1993*) and orientation of the cuttings on the parent-plant (studied in *Dalbergia melanoxylon*: *Amri et al., 2010*; *Tectona grandis*: *Husen & Pal, 2007b*; *Dalbergia sissoo*: *Husen, 2004*). This effect on rooting is due to the influence these two factors have on the availability of internally stored resources such as endogenous growth hormones and carbohydrates (soluble sugars and starch). Stored carbohydrates provide the energy

required for cell division and, thus, are required for root formation and growth (*Husen & Pal, 2007a*). Furthermore, growth hormones such as auxins play a role in the metabolism and mobilisation of carbohydrates (*Ruedell, De Almeida & Fett-Neto, 2015*; *Husen, 2008*). A lack of either of these resources can limit the rooting potential of cuttings (*Amri et al., 2010*). Older plant tissue tends to exhibit reduced carbohydrate content (*Husen, 2008*; *Husen & Pal, 2007a*), a loss of sensitivity to growth hormones, decreased endogenous growth hormone content, and an accumulation of growth inhibitory substances (*Bidabadi & Jain, 2020*; *Zargar & Kumar, 2018*; *Ikeuchi et al., 2016*; *Amri et al., 2010*; *Husen & Pal, 2007a*). Similarly, the source location of cuttings from their parent plant can influence carbohydrate content and sensitivity to growth hormones (*Amri et al., 2010*; *Husen & Pal, 2007b*; *Husen, 2004*; *Zalesny et al., 2003*). These studies show that cuttings taken from different positions (apical, basal, or mid stem) therefore exhibit different rooting potentials. The position in which to source cuttings, however, appears to be species-specific and was not consistent across the studies cited here. Tissue age and cutting location were not considered while harvesting *P. afra* cuttings, nor was the overall state of the plant or its recent history; however, these factors may have affected rooting at the parent-plant level. Each individuals' cuttings were sourced from a range of branches (3-5) that were harvested and later split into multiple cuttings of similar size for propagation. Therefore, the cuttings from an individual may have varied in relative position from the parent-plant and age. All the plants were harvested from the same macro-environmental conditions, *i.e.,* from the same slope and within 20 m of each other. Currently, the effect of the parent source material on the cutting establishment and survival has not been tested in *P. afra* restoration, but this should be explored at the intra- and inter-plant level to inform future harvesting practices for thicket restoration.

## Implications for thicket restoration

Succulent thicket restoration currently utilises unrooted *P. afra* cuttings that are harvested indiscriminately and planted with limited regard for the wetter spring and autumn months (*Mills et al., 2015*). This approach has resulted in variable success, with survival ranging between zero % and almost 100% (mean survival estimated to be 28%) in large scale plantings (*Mills & Robson, 2017*). The low survival reported has since been attributed to uncontrolled herbivory and planting into the incorrect target habitat (van der Vyver et al., 2021b), with a large number of experimental plots established in adjacent frost-prone shrublands (*Duker et al., 2020*) that cannot support the frost sensitive *Portulacaria afra* (*Duker et al., 2015*).

The potential effects of rainfall after planting and *P. afra* source material have not previously been considered in the evaluation of factors influencing cutting survival under field conditions. While watering at planting was included in the TWP experiment, it was not found to have a significant impact on *P. afra* cutting survival (*Van der Vyver, Mills & Cowling, 2021a*). Our results suggest that a lack of water, at least for the first 28 days, may have only a minor effect on rooting success, and thus establishment rates. The results from *Van der Vyver, Mills & Cowling, 2021a* and from this study suggest that watering at the time of planting (which can add considerable expense to the planting operation: *Mills et*

*al., 2018*) may not be necessary; however, the rooting window needs to be assessed beyond a waterless period of 28 days. The lack of a watering effect reported by *Van der Vyver, Mills & Cowling (2021a)* may be a consequence of rainfall soon after planting, nullifying the treatment effect, or long periods (multiple months) of no rainfall post-planting, which are common in succulent thicket habitats (*Mahlalela et al., 2020*; *Palmer et al., 2020*; *Vlok, Euston-Brown & Cowling, 2003*). Furthermore, if material is not randomised, the use of cuttings harvested from poor parent material (*i.e.,* harvesting from a nearby, old, or physiologically stressed *P. afra* population) may impact the survival of entire experimental plots or treatments within plots, as demonstrated in the significant parent-plant effects on rooting reported here. Thus, further complicating the interpretation of the TWP results.

Further work is required to both better understand the rooting window and describe the processes underpinning the parent-plant effect and the water requirements for initiating rooting in *P. afra* as this information may inform best practices for future restoration initiatives. The maximum period without water tested in this experiment—28 days—was insufficient to detect notable declines in rooting and establishment. We suggest that further experimentation with far longer periods—*e.g.,* repeating a similar experiment but staggered water treatments with a maximum of four to six months without water—may improve our understanding of the rooting window. Personal field observations (AJP) of material planted during a prolonged seasonal drought—where unrooted cuttings experienced approximately five months with no rainfall—were able to root when dug up (with no signs of rooting) and provided water in a nursery setting ($n = 15$). In addition, the results presented here suggest that soil moisture is required to facilitate root formation, as the waterless control exhibited no root development, but source material may have a greater impact on *P. afra* cutting establishment. Planting efforts should possibly be timed to best take advantage of the rainfall patterns specific to the restoration site in question, and seasonal planting of cuttings sourced from the best available parent material during the wetter autumn and spring months may be advised; however, there can be substantial inter-annual variation in these bimodal rainfall peaks.

## CONCLUSION

The restoration of succulent thicket, a dwarf forest vegetation endemic to South Africa, is dependent on better understanding the regeneration dynamics of the primary target plant species, *P. afra*. This study provides evidence that the timing of water exposure post planting of stem cuttings does have a minor impact the initiation of root development. However, unexpectedly, stronger evidence for a parent-plant effect on root development was detected. These results highlight the complexity of interpreting the drivers of *P. afra* survival under field conditions. Further work is required to better understand the parent-plant effect documented here and to inform restoration practices.

### Funding

National Research Foundation of South Africa: 119379. Nelson Mandela Universities' Postdoctoral Research Fellow Grant Program. Natural Resource Management Programme of the South African Department of Forestry. Fisheries and the Environment: E1406. The funders had no role in study design, data collection and analysis, decision to publish, or preparation of the manuscript.

### Grant Disclosures

The following grant information was disclosed by the authors:
National Research Foundation of South Africa: 119379.
Nelson Mandela Universities' Postdoctoral Research Fellow Grant Program.
Natural Resource Management Programme of the South African Department of Forestry.
Fisheries and the Environment: E1406.

### Competing Interests

Alastair J. Potts is an Academic Editor for PeerJ.

### Author Contributions

- Nicholas C. Galuszynski conceived and designed the experiments, performed the experiments, analyzed the data, prepared figures and/or tables, authored or reviewed drafts of the article, and approved the final draft.
- Ryan E. Forbes conceived and designed the experiments, performed the experiments, analyzed the data, prepared figures and/or tables, authored or reviewed drafts of the article, and approved the final draft.
- Gavin M. Rishworth conceived and designed the experiments, performed the experiments, analyzed the data, prepared figures and/or tables, authored or reviewed drafts of the article, and approved the final draft.
- Alastair J. Potts conceived and designed the experiments, performed the experiments, analyzed the data, prepared figures and/or tables, authored or reviewed drafts of the article, and approved the final draft.

### Data Availability

    The raw data and R-script used for analysis are available in the Supplementary File.

### Supplemental Information

Supplemental information for this article can be found online at http://dx.doi.org/10.7717/peerj.15538#supplemental-information.

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
