# Peer review of "Restoring South African subtropical succulent thicket using Portulacaria afra: exploring the rooting window hypothesis"

_PeerJ, doi:10.7717/peerj.15538_

## Round 0.1 · original submission · Minor Revisions

Both reviewers appreciated this work and both indicated only minor revisions are required. I agree. I encourage the authors to seriously consider the suggestions of both reviewers about slight changes to the discussion given the weight of the results. Note that reviewer 2 provided annotated comments -- many are editorial and I leave it to you to decide which to address and which to ignore; addressing all minor editorial comments is not required for final acceptance.

From my own reading, I have a few suggestions/questions, too:

1. The analysis contains both hypothesis testing based (analyzing experimental results as set out a priori) and data exploration (analyzing the parent plant interaction, which does not seem to be an a priori part of the experiment). I encourage the authors to more clearly separate these two analyses and to identify them as hypothesis testing and data exploration very clearly. This could also go a long way toward making it more obvious why a rather complicated statistical analysis was done.

2. I noticed some minor formatting issues. Not sure if these are due to the submission software or not. But, the reports of H and F statistics should have the degrees of freedom, etc. numbers as subscripts.

3. Table 1 should used periods (.) rather than commas (,) to denote decimal points (there is currently a mixture). Sorry about this, as I know that many countries use commas; but journal style is to use periods for decimal points. Thanks for your understanding.

4. Figure 1 is a little confusing. Are the grey bars the range? Is each point a parent plant? There are 7 parent plant labels for D01 but only three points -- are they overlapping? Please make this figure clearer with either a more explanatory caption and/or revised graphics.

5. I tend to agree with the second reviewer that there might need to be a larger focus on the fact that watering treatment seemed to not matter (e.g., the "All" panel in Figure 2). This has important implications, as mentioned by reviewer 2. Please seriously consider these comments.

Reviewer 1 ·

Basic reporting

No comment

Experimental design

no comment

Validity of the findings

no comment

Additional comments

Very relevant research and generally a well carried out experiment, with interesting findings. I was especially interested by the section of the discussion around the difference in rooting between earlier onset vs. later onset of watering (lines 234-237). Good use of statistical analyses in order to show trends not readily evident otherwise as being significant. I would however have expected to see the experiment carried on for a longer period of time in order to determine the longer-term variation in rooting (perhaps 3-4 months), and/or would have lagged the time-staggering to a greater window (at least 2 weeks rather than one week intervals between onset of watering, and perhaps also a longer time period (eg. 2 to 3 months) before the onset of watering to try and determine the rooting window more effectively. This is a very hardy species and cuttings will likely be able to persist at least under certain conditions before no longer having sufficient resources to begin rooting. Perhaps this was not viable within the limitations of the study, but would be an interesting follow-on study to investigate. Nevertheless, the study in the present form contributes valuable findings which have important practical implications for restoration of the species and ecosystem in which it occurs.

·

Basic reporting

The study is simple and well-designed in principle, although the short duration of the experiment does hamper the key conclusion of the ‘rooting window’.

The figures and tables are well laid out and understandable.

The results do not allow the authors to make any firm conclusions regarding the hypotheses, which is unfortunate.

I have used Word tracked-changes to indicate editorial issues (attached).

Experimental design

The experimental design is simple and well designed, and allows what seems to be the appropriate stats (apologies for my lack of comment on the stats).

Validity of the findings

The findings are valid in themselves, although I tend to think the key result is the lack of significant rooting pattern that comes from the treatment effect. It would appear to me that the statement that water is NOT needed for rooting success (within the time frame of this experiment) gives hope to the massive restoration projects that cannot provide water at planting or thereafter. In this regard, I suggest some of the key statements in the discussion and conclusion could be re-articulated to reflect this, as opposed to saying that early watering does promote rooting.

Additional comments

I found the discussion section that tries to explain a (low significance) pattern of rooting response to watering at a hormonal level a bit too speculative. I think there may be truth in the speculation, but the authors quote many unrelated studies that are not easily comparable, and no conclusion can actually be drawn from the section. I would trim this a bit.

The incidental result regarding the parent material is such an important result as it COULD offer the key to understanding the variable success of previous field plantings. I would more strongly emphasise this aspect.

---

## Round 0.2 · Minor Revisions

Thank you for addressing all the reviewer comments with such detail. I am ready to accept this manuscript but want to suggest two very minor edits before doing so. Apologies I missed these the first time:

1. I believe it is Kruskal-Wallis, not Kruskal Wallace.
2. Is the Kruskal-Wallis statistic being reported the H-value or the Chi-square? The text suggests it is the H-value, but standard R techniques typically return a Chi-square. Please confirm.

That's it! Once these two items are addressed/confirmed I will rapidly accept the paper for publication.

---

## Round 0.3 · accepted · Accept

Thank you for your diligence in responding to the final few editorial comments. It is my pleasure to accept this manuscript.